# Targeting CDK9 for Anti-Cancer Therapeutics

**DOI:** 10.3390/cancers13092181

**Published:** 2021-05-01

**Authors:** Ranadip Mandal, Sven Becker, Klaus Strebhardt

**Affiliations:** 1Department of Gynecology and Obstetrics, Johann Wolfgang Goethe University, Theodor-Stern-Kai 7, 60590 Frankfurt am Main, Germany; ranadip.mandal@kgu.de (R.M.); sven.becker@kgu.de (S.B.); 2German Cancer Consortium (DKTK), 69120 Heidelberg, Germany

**Keywords:** CDK9, Cyclin T1, RNAP II, Transcription, BRD4, MYC, Apoptosis

## Abstract

**Simple Summary:**

CDK9, in combination with Cyclin T1, is one of the major regulators of RNA Polymerase II mediated productive transcription of critical genes in any cell. The activity of CDK9 is significantly up-regulated in a wide variety of cancer entities, to aid in the overexpression of genes responsible for the regulation of functions, which are beneficial to the cancer cells, like proliferation, survival, cell cycle regulation, DNA damage repair and metastasis. Enhanced CDK9 activity, therefore, leads to poorer prognosis in many cancer types, offering the rationale to target it using small-molecule inhibitors. Several, increasingly specific inhibitors, have been developed, some of which are presently in clinical trials. Other approaches being tested involve combining inhibitors against CDK9 activity with those against CDK9’s upstream regulators like BRD4, SEC and HSP90; or downstream effectors like cMYC and MCL-1. The inhibition of CDK9’s activity holds the potential to be a highly effective anti-cancer therapeutic.

**Abstract:**

Cyclin Dependent Kinase 9 (CDK9) is one of the most important transcription regulatory members of the CDK family. In conjunction with its main cyclin partner—Cyclin T1, it forms the Positive Transcription Elongation Factor b (P-TEFb) whose primary function in eukaryotic cells is to mediate the positive transcription elongation of nascent mRNA strands, by phosphorylating the S2 residues of the YSPTSPS tandem repeats at the C-terminus domain (CTD) of RNA Polymerase II (RNAP II). To aid in this process, P-TEFb also simultaneously phosphorylates and inactivates a number of negative transcription regulators like 5,6-dichloro-1-β-D-ribofuranosylbenzimidazole (DRB) Sensitivity-Inducing Factor (DSIF) and Negative Elongation Factor (NELF). Significantly enhanced activity of CDK9 is observed in multiple cancer types, which is universally associated with significantly shortened Overall Survival (OS) of the patients. In these cancer types, CDK9 regulates a plethora of cellular functions including proliferation, survival, cell cycle regulation, DNA damage repair and metastasis. Due to the extremely critical role of CDK9 in cancer cells, inhibiting its functions has been the subject of intense research, resulting the development of multiple, increasingly specific small-molecule inhibitors, some of which are presently in clinical trials. The search for newer generation CDK9 inhibitors with higher specificity and lower potential toxicities and suitable combination therapies continues. In fact, the Phase I clinical trials of the latest, highly specific CDK9 inhibitor BAY1251152, against different solid tumors have shown good anti-tumor and on-target activities and pharmacokinetics, combined with manageable safety profile while the phase I and II clinical trials of another inhibitor AT-7519 have been undertaken or are undergoing. To enhance the effectiveness and target diversity and reduce potential drug-resistance, the future of CDK9 inhibition would likely involve combining CDK9 inhibitors with inhibitors like those against BRD4, SEC, MYC, MCL-1 and HSP90.

## 1. Cyclin Dependent Kinases (CDKs)

Cyclin Dependent Kinases (CDKs) are a family of serine/threonine kinases which require a regulatory cyclin (with the exception of CDK5, which require p35/p39) subunit to attain their kinase activity. Presently, this family comprises of 21 members, further classified into three sub-types, broadly based on their regulatory functions: (1) regulators of the cell-cycle—CDKs 1, 2, 4 and 6; (2) regulators of transcription—CDKs 7, 8, 9, 12, 13 and 19; (3) regulators of diverse or as yet undefined functions—CDKs 5, 10, 11, 14, 15, 16, 17, 18 and 20 [1,2,3]. At present, 29 cyclins have been identified in humans. While, some CDKs can have multiple cyclin partners, some cyclins can also partner with multiple CDKs. There are still some orphan CDKs whose cyclin partners have not been identified (Table 1). This review article is primarily going to focus on CDK9.

## 2. CDK9

### 2.1. The Structure of CDK9

CDK9 was originally discovered by Graña X. et.al. in 1994, in the quest to identify potential CDC2 (CDK1) related kinases, as a 42 kDa, 372 amino acid (aa) protein, termed PITALRE due to the presence of a Pro-Ile-Thr-Ala-Lue-Arg-Glu containing motif [29]. This motif corresponded with the highly conserved PSTAIRE box, seen in multiple CDKs (Figure 1 and Figure 2A) [30].

CDK9 exists in two isoforms, the originally identified and more abundant one of 42 KDa and the less abundant one of 55 kDa, the latter having an additional 117 aa at its N-terminus [12]. These two isoforms are transcribed from the same *CDK9* gene, but by two different promoters, located more than 500 bp apart on the *CDK9* gene, with the 42 kDa promoter being significantly stronger than the 55 kDa one [31]. Within the nucleus, CDK9_42_ is primarily localized within the nucleoplasm while CDK9_55_ primarily accumulates within the nucleolus [12] (Figure 2A). All amino acid positions and other aspects of CDK9 mentioned henceforth would be about CDK9_42_.

### 2.2. The Activation of CDK9

Unlike many CDKs, the activation of CDK9 is not regulated in a cell-cycle dependent manner but by its association with its cyclin partners to form a heterodimeric complex Positive-Transcription Elongation Factor-b (P-TEFb). Even though Cyclin T1 is the primary cyclin partner of CDK9, minor partners like Cyclins T2a and T2b can also activate CDK9 [1,32,33]. In fact, in HeLa cells, the shRNA mediated down-regulation of Cyclin T1 led to the down-regulation of 631 genes, as compared to just 292 genes down-regulated upon the knock-down of Cyclin T2 (both Cyclin T2a and Cyclin T2b) [33] (Figure 2B). Initially, Cyclin K was also considered to be another activator of CDK9 [34], however, it is now considered to be a primary activating partner of CDKs 12 and 13 [35,36].

The activity and substrate specificity of P-TEFb are decided by the phosphorylation of a number of serine and threonine residues like—T29 and S90 at the N-terminus; S347, T350, S 353, T354, S357, S362 and T363 at the C-terminus; S175 and T186 at the T-loop of CDK9 [30,37,38,39,40,41] (Figure 2A). Amongst them, the phosphorylation of T186 (pT186) is essential for the kinase activity of CDK9 and association with Cyclin T1, as evidenced by the fact that T186A and T186D mutants reduced this association by ~90% and ~50%, respectively, as compared to WT-CDK9 (CDK9_WT_) [39]. The analysis of the X-ray crystallographic structure of P-TEFb revealed that T186 was in close proximity and interacts with the arginine residues R148 and R172 [32] and R65 [38] (Figure 2A), to form an intra-molecular H-bond network. T186 was also reported to coordinate and stabilize an inter-molecular salt-bridge between R65 and R172 of CDK9 and E96 of Cyclin T1 (Figure 2B). Like T186A [39], mutations at these three arginine residues (R65A, R148A and R172A) resulted in an almost complete loss of interaction with Cyclin T1, while R65A and R172A also led to a loss of pT186, suggesting a mutual stabilization mechanism. On the other hand, E96A and E96K mutations of Cyclin T1 caused a partial and complete loss of P-TEFb formation, respectively [38].

Initially, it was reported that, when synthesized, CDK9 undergoes auto-phosphorylation at the T186 residue on its T-loop. When Kim et al. had expressed a mutated CDK9 (T186A) in insect cells using the Baculovirus expression system, no phosphorylation could be detected, even in the presence of Cyclin T1, although, the kinase activity of the mutant remained unimpeded. Incubating CDK9–WT; T186A; and D167N (kinase dead) with the CDK Activating Kinase (CAK; CDK7/Cyclin H), showed a phosphorylation signal only in the CDK9_WT_ but in none of the mutants. Even in the presence of Cyclin T1, CAK couldn’t enhance the phosphorylation signal of CDK9 above the background level, allowing them to conclude that CDK9 underwent auto-phosphorylation, irrespective of the presence of Cyclin T1 and this phosphorylation was not critical for the kinase activity of CDK9 [42]. CDK9 T186 auto-phosphorylation was also confirmed by Baumli et al. following the mass spectrometric analysis of full-length CDK9, in complex with Cyclin T1 [32].

Larochelle et al. had demonstrated that insect cell-derived CDK9_D167N_, both as monomer or in combination with Cyclin T1, purified from bacteria, were phosphorylated at T186 [37], partially contradicting the earlier observation [42]. Similarly, generated CDK9_WT_ also showed pT186 and could not be further activated by CAK. However, unphosphorylated WT CDK9, synthesized in vitro by programming rabbit reticulocyte lysates, could not be phosphorylated at T186, even in the presence of Cyclin T1 and ATP, but elicited basal activity towards SPT5. When CAK was introduced into the reaction, pT186 appeared as well as a great enhancement in SPT5 phosphorylation. Similar trend in T186 phosphorylation was also seen with CDK9_D167N_, albeit at much lower levels and no activity towards SPT5 (Figure 2C) (explained in the following sections). Introducing CAK to this reaction, phosphorylated WT CDK9 at T186 and increased its activity towards SPT5. The CDK9_D167N_ was also phosphorylated at T186, albeit at much lower level and had no kinase activity, proving that Cyclin T1 is critical for the activity of CDK9. In HCT116 cell lines, CAK was also shown to be responsible for almost all of pT186, as well as the activation of CDK9 on transcribed chromatin. Additionally, the selective inhibition of CDK7 diminished CDK9 mediated pS2 of RNAP II (Figure 2G) [37].

Furthermore, an siRNA library screen of 78 ubiquitously expressed serine/threonine kinases, to identify potential CDK9 T-loop kinases, had revealed the Ca^2+^/Calmodulin-dependent Kinase 1D (CaMK1D) as one such target. Interestingly, although the siRNA mediated knock-down of CaMK1D led to the reduction in pT186, this effect could not be reversed neither by the overexpression of ectopic CaMK1D, nor by increasing Ca^2+^ levels in HeLa cells. CDK9 and CaMK1D were also not found to interact in either the cytosol or nucleus of HeLa cells, indicating that CaMK1D indirectly affects the CDK9 pT186. In contrast, inhibiting CaMK1D activity by the CaMK inhibitor KN-93 or the CaM inhibitor W-7, led to 59% and 29% reduction in pT186 levels, respectively, in HeLa cells. A key takeaway from this work was the revelation that the knock-down of expression or inhibition of activity of CaMK1D also led to the corresponding reduction in total CDK9 levels. This reduction was revealed to be due to the proteasomal degradation of CDK9, which could be partially reversed by treating HeLa cells for 1 h with the protease inhibitor MG-101 [41].

### 2.3. RNA Polymerase II-Mediated Transcription

The molecular function of CDK9 cannot be described without briefly explaining transcription. In human cells, the expressions of many protein coding genes are regulated by the RNA Polymerase II (RNAP II), at several steps like pre-initiation, initiation, elongation, RNA processing and termination. For the sake of this review, we will briefly focus on some of these steps. Transcription initiation by RNAP II begins with the formation of the Pre-Initiation Complex (PIC) (Figure 3A and Figure 4A–C), for which RNAP II is assisted by General Transcription Factors (GTFs) like TFIIA, TFIIB, TFIID, TFIIE, TFIIF and TFIIH. TFIID in-turn is composed of the TATA box-Binding Protein (TBP), which is required for the transcription from every promoter, and several TBP Associated Factors (TAFs) (Figure 3A), which are required for promoter specific transcription [43,44,45].

TFIIH is a multi-protein complex, comprising of, in its transcription initiator role—10 proteins (XPB, XPD, p62, p52, p44, p34 and p8 forming the core, plus CAK with CDK7, Cyclin H and MAT1) [46] (Figure 3B). After the formation of the PIC, the transcription of a gene initiates subsequent to the TFIIH mediated—(1) phosphorylation of RNAP II, at the S5 (pS5) residues of the Y_1_S_2_P_3_T_4_S_5_P_6_S_7_ tandem repeats and (2) ‘promoter melting’ which involves opening-up ~10 bp of the promoter dsDNA, allowing RNAP II to access the template strand [43,47,48]. The human RNAP II CTD possesses 52 of this heptad repeats [49]. The pS5 also allows the removal of the CDK8 containing Mediator Complex (MC) (Figure 3D and Figure 4D) and the release of RNAP II from the PIC [45]. During transcription initiation, RNAP II generates a nascent transcript ~50 ribonucleotides from the promoter and then pauses due to the combined effect of two negative regulators—(1) NELF and (2) DSIF (Figure 4D) [50]. One final consequence of the pS5 is the recruitment and activation of capping enzymes which perform a multi-step 5’-capping of the nascent mRNA strand (Figure 4D). This capping prevents the degradation of the nascent mRNA strand by nucleases like XRN2 (5′-3′ exoribonuclease 2) and the resulting termination of the promoter proximally paused RNAP II [45,51,52]. The next step in RNAP II mediated transcription is transcriptional elongation by P-TEFb. Interestingly, the regulation of the expressions of most genes in human cells occurs during their transcription elongation, rather than at their initiation phase [45,53,54]. Thus, CDK7 aids in both pausing transcription and relieving it by activating CDK9 [38] (Figure 4).

### 2.4. The Molecular Functions of CDK9

A great deal of the initial information about the function of P-TEFb was obtained by studying its function in HIV-1 transcription, which is regulated by the viral Trans-Activator protein Tat. In the absence of Tat, RNAP II undergoes promoter proximal pausing, in a region adjacent to the Trans-Activation Response element (TAR) RNA sequence, to synthesize short transcripts of ~60 ribonucleotides [38,55]. When present, Tat promotes HIV-1 transcription by first binding to the bulge of the TAR hairpin-loop and then recruiting P-TEFb, in complex with the Super Elongation Complex (SEC) (Figure 3E), to the HIV-1 promoter. When P-TEFb was knocked-down by siRNA in HeLa cells, it did not cause the cell death but inhibited Tat activation and HIV-1 transcription [47,56]. The previously mentioned phosphorylation sites in CDK9-S347, T350, S353, S357 and T354 were found to be essential for the binding of P-TEFb to TAR, as P-TEFb with a C-terminal truncated CDK9 (Δ323), bound to TAR significantly weakly, as compared to CDK9_WT_ [40,57]. In fact, the then newly found protein was called Cyclin T due to its unique ability to bind with Tat [58]. After two additional Cyclin Ts (T2a and T2b), which cannot interact with Tat, were discovered, the original was termed Cyclin T1 [59].

As mentioned in the previous section, DSIF and NELF are responsible for the promoter proximal pausing of RNAP II (Figure 4F). P-TEFb plays an essential role in promoting transcription elongation by performing three functions that release RNAP II from its pause—(1) phosphorylating NELF, causing its dissociation from the paused RNAP II, (2) phosphorylating the SPT5 subunit of DSIF (Figure 2C and Figure 4F), at T4 of its CTD GS(Q/R)TP residue [60], which subsequent to NELF dissociation, converts into a positive elongation factor and (3) phosphorylating RNAP II at the S2 (pS2) residues of the Y_1_S_2_P_3_T_4_S_5_P_6_S_7_ repeats at its CTD. Following the release of RNAP II from its promoter proximal pause, the kinase activity of P-TEFb is rendered unnecessary, although it still travels along the Transcription Elongation Complex (TEC) [38,40,50,61].

### 2.5. The Regulation of CDK9 Activity

In order to regulate a highly choreographed process like transcription, the activation and activity of transcription are tightly regulated. In the nuclear extract of HeLa cells, ~80% of CDK9 was found to be associated with Cyclin T1 (P-TEFb) and 10% each with Cyclins T2a and T2b [59]. Of these, ~50% of P-TEFb is sequestered in the 7SK snRNP (small nuclear Ribonuclear Protein) [50,66,67] (Figure 3C), which negatively regulates its kinase activity. The 7SK snRNP is composed of the RNAP III transcribed, non-coding 7SK snRNA, which serves as the scaffold for the RNA binding proteins HEXIM1/HEXIM2 (Hexamethylene bis-acetamide inducible proteins) (Figure 2D), LARP7 (La-Related Protein) (Figure 2E) and MePCE (Methylphosphatase Capping Enzyme) [30,50,68]. While HEXIM1 inhibits P-TEFb’s kinase activity in a 7SK dependent manner [69], LARP7 stabilizes the 7SK snRNA and protects it from exonuclease meditated degradation by binding to its 3′-UUUU-OH sequence, as evident from the fact that the knock-down of LARP7 resulted in the complete degradation of 7SK snRNA [70,71]. MePCE was originally thought to add a unique γ-monomethyl phosphate cap to the 5′-end of 7SK snRNA [72], however, recent work had shown that this function existed only in a LARP7 free environment. The presence of LARP7, on one hand, inhibited the capping activity of MePCE, but on the other hand, promoted its interaction with LARP7, to co-operatively stabilize 7SK snRNA [73]. Here too, the CDK9 pT186 played a critical role as CDK9_T186A_ or _T186D_ (Figure 2A) inhibited the associations of the resulting CDK9s to HEXIM1 and LARP7 by ~99%, as compared to CDK9_WT_, in Jurkat cells [39] as well as in HeLa cells [74]. Similarly, CDK9_R65A_, _R148A_ and _R172A_ and Cyclin T1_E96K_ (Figure 2B) were completely incapable of interacting with HEXIM1 and LARP7 [38].

The remaining 50% P-TEFb was associated with BRD4 (Figure 2F) or with the SEC (Figure 3E). BRD4 is the only member of the Bromodomain and Extraterminal (BET) protein family capable of binding with P-TEFb [30,75]. Unlike the inactive P-TEFb in the 7SK snRNP complex, P-TEFb associated with BRD4 was active and was recruited to various promoters via BRD4’s interactions with acetylated histones-H3, H4 and MC [76,77]. Interestingly, P-TEFb remained associated with BRD4 since the formation of PIC until the initiation of transcription elongation [30]. Under physiological conditions, the EAF1 and 2 of SEC (Figure 3E) interacted with the med26 subunit of the MC (Figure 3D) to get directly recruited to genes like *MYC* and *HSP70*. ChIP-seq analysis following med26 depletion resulted in lower occupancy of SEC components throughout the complete transcribed regions of the aforementioned two genes, as well as reduced levels of S2 RNAP II on these genes. However, with some genes, SEC recruitment required the pre-occupancy of BRD4, while in others, the RNAP II associated factor (PAF1) recruited SEC by interacting with AF9 or ENL [64,65].

Multiple hypothesis had been put forth to explain the interaction between BRD4 and P-TEFb. Zhou et al. had reported that BRD4, bound to P-TEFb, was recruited to HIV PIC (Figure 3A and Figure 4E) where, in order to subsequently facilitate the formation of an efficient TEC, transiently inhibited the kinase activity of P-TEFb by auto-phosphorylating the T29 of CDK9. As the PIC gears-up for transcription elongation, between positions +1–+14 for HIV-1, BRD4 was released from the TEC. The phosphatase PP2α was then recruited to the TEC, which de-phosphorylated T29, restoring the kinase activity of P-TEFb [78]. Schröder et al. had proposed a ‘two-pronged’ binding mechanism between BRD4 and P-TEFb, where on one hand, the N-terminal Bromodomain (BD2) of BRD4 (Figure 2F) interacted with at least 3 of the 4 known acetylated lysine residues of Cyclin T1 (K380ac, K386ac and K390ac), while P-TEFb was still bound to HEXIM1 and 7SK snRNA. On the other hand, the C-terminal 54 aa long P-TEFb Interacting Domain (PID) of BRD4 (Figure 2F) interacted with the active P-TEFb, devoid of both HEXIM1 and 7SK snRNA, by dissociating HEXIM1 from Cyclin T1. This marked the transition between inactive and active P-TEFb in the presence of BRD4. Of note, BD1 showed no binding affinity towards Cyclin T1 and the affinity of BD2 towards the tri-acetylated Cyclin T1 was similar to its affinity towards acetylated histones [79]. This finding was further corroborated by Itzen et al. who had demonstrated that in vitro, PID stimulated the kinase activity of P-TEFb by ~1.3 folds over its basal activity, even in the presence of HEXIM1 (Figure 2D). By performing Isothermal Titration Calorimetry, they had also demonstrated that PID did not bind to Cyclin T1, in the absence of CDK9, thereby proposing that the Bromodomain and PID interacted synergistically with Cyclin T1 and CDK9 of P-TEFb, respectively, forming a stable clamp ~900 aa apart. Interestingly, PID-P-TEFb interaction did not alter the phosphorylation specificity and substrate preference for CDK9 at the CTD of RNAP II [80]. Yang et al. had also reported the importance of a new CDK9 phosphorylation site–S175, in the biding of P-TEFb with BRD4. When mutated to CDK9_S175A/D_ in HeLa cells, the association of the mutant CDK9 with BRD4 was completely abolished, without affecting the association with either Cyclin T1 or HEXIM1, indicating that the pS175 is necessary for the association of P-TEFb with BRD4 and its resulting dissociation from the inactive 7SK snRNP complex [74,81]. This was also confirmed by Mbonye et al. in T-cells [38,39]. On the other hand, CDK9_T186A/E_ almost completely abolished the association of the mutant CDK9 with HEXIM1 but not with Cyclin T1 or BRD4, confirming the need for this phospho-site for HEXIM1 binding [81,82], although Mbonye et al. had shown that CDK9_T186A_ couldn’t interact with Cyclin T1 [38]. The kinase responsible for the pS175 was eventually identified as CDK7. Treatment of Jurkat cells with PMA led to the partial dissociation of P-TEFb from the 7SK snRNP (Figure 3C) and therefore pS175 level, which was effectively inhibited by the selective CDK7 inhibitor THZ1. pT186 remained unaffected [38].

From all these and other works, it has become evident that BRD4 and other promoter-specific TFs are necessary to retrieve the active P-TEFb from the 7SK snRNP complex and carry out transcription elongation.

### 2.6. The Other Pool of CDK9

As mentioned earlier, ~80% of CDK9 existed as P-TEFb, whether sequestered as an inactive complex with 7SK snRNP or as an active complex with BRD4 or other TFs. But, what about the remaining 20% CDK9, unbound to Cyclin T1? O’Keeffe et al. had demonstrated that free CDK9 was unstable within the cells and got degraded rapidly. In 293T cells, stably expressing a HA-tagged CDK9, ~80% of the newly synthesized HA-CDK9 was degraded within the first 6–12 h, whereas the remaining ~20% HA-CDK9 did not show any significant degradation until 48 h. They were the first to identify that CDK9 bound to two chaperones—the more general HSP70 and the substrate specific HSP90/CDC37 complex, which acted in a sequential manner. Initially, HSP70 stabilized any newly synthesized CDK9 by correctly folding it. Once the nascent CDK9 attained a certain folded configuration, it was handed over to the HSP90/CDC37 complex, which established and maintained the CDK9 for the follow-up binding to Cyclin T1, thereby completing the final assembly of P-TEFb. The inhibition of HSP90’s interaction potential by the inhibitor Geldanamycin led to ~12.5 folds increase in HSP70 bound HA-CDK9. The erstwhile mentioned stable ~20% CDK9 comprised of these three complexes [83]. More recently, Mbonye et al. had confirmed these findings in T-cells. They had also described that in the absence of Cyclin T1 expression, as in resting T-cells, HSP90/CDC37 bound CDK9 was predominantly localized within the cytosol and lacked any pT186. Upon undergoing TCR activation, resulting in Cyclin T1 expression, CDK9 was imported to the nucleus to get activated. Once again, the treatment of Th17 cells with the Geldanamycin derivative-17-AAG significantly reduced the formation of P-TEFb and the expression of Cyclin T1, possibly through proteasomal its degradation. Additionally, the association of CDK9 with HSP90/CDC37 showed a minor, but significant, uptick in CDK9_T186A_ [38]. Indeed, Napolitano et al., had demonstrated that CDK9 could be detected in both the cytosol and the nucleus, while Cyclin T1 was predominantly localized within the nucleus. Our own work in this direction has also confirmed this in multiple cervical, breast and ovarian cancer cell lines (un-published work) [84,85]. It was also shown that CDK9 could shuttle between the nucleus and cytosol, via the CRM1/exportin protein and inhibiting the association between CDK9 and CRM1 by the nuclear export inhibitor LMB (Leptomycin B) enriched nuclear CDK9. Additionally, it was demonstrated that the expression of Cyclin T1 promoted the nuclear localization of CDK9 [86,87].

## 3. The Clinical Relevance of CDK9

CDK9 expression had been implicated in the prognosis and resistance to anti-cancer therapeutics in a large number of cancer types like those of breast, lung, prostate, endometrium, apart from melanoma, osteosarcoma, myeloid leukemia, soft tissue sarcomas etc. [1,88,89,90,91,92]. We will briefly explain some of these cancer entities here.

### 3.1. Breast Cancer

ERα^+^ endocrine therapy-resistant breast cancers over-expressed the cMYC oncogene and genes regulated by it [93] which correlated with poor RFS, but not when the patients had only undergone chemotherapy [89]. In a study on breast cancer cell lines, Sengupta et al. had demonstrated that in multiple aromatase inhibitor-resistant MCF-7 cancer cell lines, the levels of cMYC mRNA and proteins were significantly higher than the parental MCF-7 cell line. In these cell lines, the cMYC promoter showed higher recruitment of pS2, but not pS5 in RNAP II, as compared to MCF-7 cells, which also correlated with higher levels of both pT186 and CDK9 (2.5 and 3.1 folds, respectively). Lastly, the inhibition of CDK9 activity, in the resistant cell lines, with the small-molecule inhibitor CAN-508, showed a dose-dependent inhibition of cell growth, in conjunction with ~60% reduction in cMYC mRNA and proteins, highlighting the critical role of CDK9 in the up-regulation of cMYC expression in endocrine therapy-resistant ERα^+^ breast cancer patients [89].

### 3.2. Osteosarcoma

Similarly, Ma et al. had demonstrated that, of 70 osteosarcoma patient derived samples, 67.1% exhibited high CDK9 expression, which significantly correlated with metastatic disease, non-survival, worse OS (Overall Survival) and DFS (Disease Free Survival) and poor post-neoadjuvant chemotherapy necrosis. CDK9 expression was also higher in osteosarcoma cell lines, as compared to the osteoblast cell line HOB-c. Down-regulating CDK9 (siCDK9) expression led to decreased cell proliferation, as well as reduced levels of pS2 RNAP II and MCL-1 whereas, treating the osteosarcoma cell lines with the ATP competitive CDK9 inhibitor LDC000067 significantly reduced the levels of pS2 RNAP II and MCL-1 and their clonogenicity, in a dose-dependent manner, in conjunction with increased apoptosis. 12 days of treatment with 10µM of LDC000067 also significantly reduced the diameters of the 3D spheroids of U2OS (57.4%) and KHOS (48.8%) cell lines, as compared to their untreated counterparts [88,94].

### 3.3. Endometrial Cancer

Of the 32 endometrial cancer patient samples (carcinosarcoma-2, clear cell-2, serous-4, mixed cell-2, mucinous-1 and endometrioid-21), analyzed by He et al., 59.4% had low and 40.6% had high CDK9 expression, but demonstrated no correlation between these expression values and the pathological stage, grade and sub-type of endometrial tumor. As seen before, high CDK9 expression significantly worsened the OS (39 vs. 14 months) and PFS (Progression Free Survival) (96 vs. 26 months). Concentration dependent knock-down of CDK9 expression and dose-dependent inhibition of CDK9 activity by LDC000067 led to correspondingly significant reductions in cell viability and MCL-1 expression while BAX expression and PARP cleavage were enhanced. Additionally, 14 days of treatment with increasing concentrations of LDC000067 resulted in significant reductions of clonogenicity and 2D migration of endometrial cancer cell lines [90].

### 3.4. AML

The dysregulation of the CDK9 pathway was also reported by various groups in AML. Increased expressions of cMYC and MCL-1 had been associated with the pathogenicity of AML due to the enhanced survival and expansion of the AML cells. High MCL-1 expression was also observed in and associated with poor prognosis in ~50% of R/R AML cases. Translocation products of the *MLL* (Mixed Lineage Leukemia) gene were found to associate with P-TEFb in AML to trigger constitutively active transcription [95]. Its levels were found to be ~2 folds higher in recurrent patients as compared to pre-treatment patients, while its down-regulation led to the death of both human and murine AML cells [96,97]. As a result, targeting the anti-apoptotic BCL-2 family members, to which MCL-1 belongs, had been attempted. Venetoclax is one such inhibitor which was approved by the FDA in 2020, for use against newly diagnosed AML patients of 75 years or older, in combination with Azacitidine, Decitabine or Low-Dose Cytarabine (LDAC). Unfortunately, MCL-1 up-regulation enables Venetoclax resistance and relapse. Combining Venetoclax with the CDK9 inhibitor Alvociclib had shown synergistic effect against both Venetoclax resistant and sensitive AML patients, in a pre-clinical study [97,98].

On the other hand, in primary AML samples, the mRNAs of HEXIM1 and cMYC were found to be over-expressed with mutual exclusivity [99]. HEXIM1 served as the binding partner of several other proteins, nearly half of which has been recognized for their roles in cancers. In AML, the cytoplasmic-mislocated mutant of the p53 regulator NPM (Nucleoplasmin)-NPMc^+^, was observed in ~35% of all AML patients. Both NPM and NPMc^+^ were HEXIM1 binding proteins. NPM overexpression induced the proteasomal degradation of HEXIM1, thereby freeing-up P-TEFb to regulate enhanced transcription elongation. NPMc^+^ had a disrupted NLS and an extra NES, thereby localizing the resulting protein to the cytoplasm instead of the nucleoli, as seen with NPM. NPMc^+^ sequestered HEXIM1 from its usual nuclear location to the cytoplasm, again hampering the formation of the negative regulatory 7SK snRNP complex, leading to the increased P-TEFb mediated transcription elongation [100].

### 3.5. Lung Cancer

The pro-inflammatory cytokine TNFα stimulated the expression of the Matrix Metalloproteinase-9 (MMP-9) in an NF-κB dependent manner, which promoted cancer cell migration and invasion, by degrading basement membrane, and also angiogenesis, by vascular structure remodeling and the release of angiogenic factors like VEGF (Vascular Endothelial Growth Factor) [101,102]. Shan et al. had demonstrated that in the lung adenocarcinoma cell line A549, TNFα stimulated the transcription of the MMP-9 promoter. To achieve this, TNFα enhanced the associations of both CDK9_55_ and CDK9_42_ to Cyclin T1, as well as the binding of P-TEFb to the *MMP9* gene which led to increased levels pS2 RNAP II bound to *MMP9*. The P-TEFb inhibitor DRB substantially inhibited TNFα stimulated MMP-9 mRNA expression by ~76 folds and its activity by ~80%, but not those of TGF-ß1 stimulated MMP-2. Similarly, the over-expression of a double-negative CDK9 (CDK9dn) or an siCDK9 lowered the TNFα stimulated MMP-9 activity by more than half. Additionally, CDK9 positively regulated the NF-κB response element, through which TNFα stimulated the transcription of the MMP-9 promoter. Once again, DRB as well as CDK9dn blocked the activation of NF-κB by TNFα [103]. More recently a novel highly selective CDK9 inhibitor-21e, was shown to inhibit CDK9 activity (IC_50_ 11 nM) in NSCLC cells as well as cell lines of other cancer entities. It suppressed the clonogenicity of two NSCLC cell lines while inducing apoptosis by down-regulating the expression of BCL-2 and up-regulating the pro-apoptotic protein BIM, as well as increasing Caspase-3 activation. 21e also significantly inhibited pS2 RNAP II. Inhibiting CDK9 activity also suppressed the expressions of the stemness markers in the two NSCLC cell lines-SOX2, OCT4, NANOG and KIF4, while also significantly inhibiting other stemness phenotypes like Side Population (SP) and serum-free suspension culture. Lastly, 21e also inhibited the tumor growth of a mouse xenograft model derived from the NSCLC cell line H1299, once more validating the significance of CDK9 in lung cancer [104].

### 3.6. Prostate Cancer

Androgen Receptor (AR), a steroid receptor TF for testosterone and di-hydrotestosterone, played a critical role in prostate cancer, especially in the castration resistant sub-type. AR mutations, Post-Translational Modifications (PTMs), over-expression and splice variants were some of the ways through which prostate cancers evaded Androgen Ablation Therapy (AAT) [105]. AR was compartmentalized in the cytoplasm in an inactive form, in complex with HSP90, which sequestered it from migrating into the nucleus and binding to the DNA. In the presence of suitable steroid ligands, AR underwent a conformational change and detached from the HSP90. The conformationally changed AR then underwent homo-dimerization and translocated to the nucleus, where it bound to Androgen Response Element (ARE) located at the promoters of the target genes. The phosphorylation of AR is one PTM which resulted in its altered-activation, transcriptional activity, stability, nuclear retention and cell growth. 18 such phosphorylation sites are currently known [106], among which, S81 is most frequently phosphorylated in response to androgen stimulation, by kinases like CDK1 [107,108], CDK5 [109] and CDK9 [110]. Mutating this phosphorylation site to S81A resulted in the reduced cell growth and altered the transcription of the AR regulated genes in a promoter specific manner. In vitro, CDK9 phosphorylated S81 exclusively and ex vivo, under basal conditions, WT-CDK9/Cyclin T1 could but CDK9dn/Cyclin T1 could not phosphorylate S81, but in the presence of the synthetic androgen R1881, both constructs could phosphorylate S81, albeit to a lower level for CDK9dn/Cyclin T1. siCDK9 treatment also led to a significant reduction in pS81, in the presence of R1881. Similarly, inhibiting CDK9 activity by DRB or Flavopiridol treatment reduced pS81 signal in a dose-dependent manner, in conjunction with the reduced transcription of the AR regulated early response genes—*SGK1* and *NKX3.1* [110]. Chen et al. had shown that while CDK1 mediated pS81 increased during mitosis, CDK9 performed this function during the interphase of androgen responsive LNCaP and non-responsive PC3 prostate cancer cell lines. Additionally, CDK9 mediated pS81 stabilized AR chromatin binding for transcription [107].

### 3.7. Melanoma

CDK9 has been reported to be overexpressed in many melanoma cell lines [111] while CDKs 7 and 9 were also shown to be over-expressed in multiple uveal melanoma cell lines. Using the potent CDK7/9 inhibitor—SNS-032, it was demonstrated that pS2/5/7 at RNAP II, by these two kinases, could be inhibited in a dose-dependent manner, resulting in reduced cell proliferation, migration, invasion, cancer stemness, in vivo tumor formation, increased MMP-9, but not MMP-2, expression, in dose- and time-dependent manners. Under similar conditions, increased apoptosis of the treated cells was also observed, both *ex* and in vivo, accompanied by enhanced PARP and Caspase-3 cleavages and reduced expressions of several anti-apoptotic proteins. Lastly, SNS-032 also led to reduced transcription and expressions of YAP (Yes Associated Protein) and its active form pYAP (S127). All these alterations were attributed directly to the inhibition of CDK7/9 by SNS-032 [112].

Recently, Zhang, et al. had demonstrated that the CSN6, a member of the Constitutive Photomorphogenesis 9 (COP9) Signalosome (CSN) complex family, was over-expressed in malignant melanoma cell lines and tissues, driving their proliferation, migration and invasion. CSN6 was shown to interact with CDK9, stabilizing and protecting the latter from being ubiquitinated and subsequently degraded by UBR5 (Ubiquitin Protein ligase E3 component n-recognin 5), in the malignant melanoma cell lines A375 and MV3. Mouse xenograft models of A375 cells with stable knock-down of CSN6 manifested significantly smaller tumors and CDK9 positive cells, as compared to those derived from CSN6 expressing cells. These effects of CSN6 knock-down could be significantly rescued by concurrent knock-down of UBR5, highlighting the significance of CDK9 in melanoma [113]. As CSN6 expression is up-regulated in multiple cancer entities [113,114,115], it stands to reason that this effect, demonstrated in melanoma could occur in other cancer types too.

### 3.8. Ovarian Cancer

Amongst 26 primary ovarian cancer patient samples (serous-20, squamous-1, high-grade serous-1, endometroid-1, transitional cell-1, endometroid and serous-1 and endometroid and clear cell-1), Wang et al. had shown significantly lower CDK9 expression in primary cancer tissues than patient-paired metastatic and recurrent cancer tissues. Of these, 13 patients with lower CDK9 expression had better DFS and OS, as compared to those with higher expression. No correlations were observed between CDK9 expression and the stage, grade, subtype and ascites of the ovarian cancer patients. Similar to endometrial cancer and osteosarcoma, use of siCDK9 and LDC000067 resulted in a dose-dependent inhibition of cell-viability and a decrease in pS2 of RNAP II, MCL-1 levels and an increase in BAX expression and PARP cleavage. Additionally, LDC000067 treatment led to significant reductions of clonogenicity, 2D migration and 3D spheroid formations of two ovarian cancer cell lines—SKOV-3 and OVCAR-8, once again highlighting the significance of CDK9 in the poor prognosis of ovarian cancer [116].

## 4. Inhibitors of CDK9

A number of CDK9 inhibitors have been reported over the years, including many pan-CDK inhibitors (Table 2). Cassandri et al., in their review article, had listed a number of natural CDK targeting compounds like Indirubins, isolated from several indigo producing plants. A couple of synthetic derivatives of Indirubins like 6-bromoindirubin and 6-bromoindirubin-3′-monooxime were also reported [1]. Another natural compound Rohitukine, with anti-inflammatory and anti-immunomodulatory properties, derived from the *Dysoxylum binectariferum* plant, served as the basis for one of the earliest pan-CDK inhibitor Flavopiridol [117]. In this review, we would briefly describe Flavopiridol, as it was the first CDK9 inhibitor to enter clinical trials [118], and some of the other more recent, selective inhibitors, although not all of them entered the clinical trials phase (Table 2).

### 4.1. Flavivirid

Flavopiridol (Alvocidib) (Table 2) is a semi-synthetic flavonoid alkaloid derivative of Rohitukine, discovered following an NCI based screening of 72,000 compounds against 60 human cancer cell lines [119]. Initially, Flavopiridol showed inhibitory activities against EGFR and PKA (IC_50_ 21 and 122 µM, respectively), however, during the NCI screen, it exhibited growth inhibition at IC_50_ 66 nM, ~1000 folds lower than the concentrations required for EGFR and PKA inhibition [120]. Its remarkable anti-proliferative and growth inhibitory effects, in vivo and ex vivo, respectively, were initially attributed to its activity against CDKs 1, 2, 4 and 7. Eventually, it was found to be most effective against CDK9 [121]. Flavopiridol is an ATP-competitive inhibitor, which showed promising in vitro activity against a number of cancer entities, but demonstrated only limited in vivo activity, except in hematologic cancers like MCL and CLL, resulting in the cessation of its further development [1,122]. Presently, only three trials, involving Flavopiridol, are active, against AML, advanced solid tumors and Myelodysplastic Syndromes (MDS). Dinaciclib (SCH-727965) was developed as next generation inhibitor, based on Flavopiridol. Dinaciclib inhibited CDKs 1, 2, 5 and 9 with respective in vivo IC_50_ of 3, 1, 1 and 4 nM, as compared to 3, 12, 14 and 4 nM for its predecessor. It also triggered apoptosis and arrested cell-growth in >100 tumor cell lines [123,124]. However, preliminary results from clinical trials have been reported to be less encouraging [122]. Further trials are ongoing (Table 2).

### 4.2. AZD-4573

The development of the highly selective CDK9 inhibitor AZD-4573 (Table 2) was based on another selective, oral inhibitor of the amidopyridine series—AZ-5576, which was reasonably potent against the AML cell line MV-4-11, both ex and in vivo, but exhibited lower solubility in PBS and rate of metabolism in human microsomes, limiting its ability to provide a high therapeutic dose and optimal duration of target engagement within the clinical scenario. AZD-4573 showcased excellent selectivity against CDK9 with in vitro IC_50_, measured using the Thermofisher Kinase Panel, at <4 nM, as compared to 117, 52, 23, 499, 1270, 363, 1370 and 8070 nM against CDKs 1, 2, 3, 4, 5, 6, 7 and 12, respectively. Twice daily i.p. administration (bid) of 5 and 15 mg/kg AZD-4573 resulted in 97% and 100% tumor growth inhibition, respectively, for MV-4-11 and 65% inhibition, for both doses, against Nomo-1 based AML xenograft mouse models, along with reductions in pS2, MCL-1 levels and increase in Caspase-3 cleavage. In MCF-7 cells, AZD-4573 had an IC_50_ of 14 nM against CDK9, as compared with 370 nM, 1100 and >10 µM against CDKs 1, 4/6/7 and 2/4/6, respectively. Both the mRNA and protein levels of MCL-1 were down-regulated in a dose-dependent manner, without affecting the same for other anti-apoptotic proteins like BCL-2 and BCL-XL. Caspase-3 cleavage was also observed in MV-4-11 and MOLM-8 cell lines, suggesting that MCL-1 is the major initiator of AZD-4573 driven apoptosis. Against a diverse panel of cancer cell lines, 6 and 24 h of AZD-4573 treatment generated significant Caspase-3 activation and reduction in viability, more pronouncedly in hematological cancer cell lines than in solid tumor cell lines. The pharmacokinetics (PK) and pharmacodynamics (PD) of AZD-4573 also improved in vivo over AZ-5576 [125,126]. Presently, AZD-4573 is subject to two phase-I clinical trials against advanced hematological cancers (NCT04630756) [127] and relapsed/refractory hematological cancers (NCT03263637) [128] (Table 2).

### 4.3. BAY-1143572 (Atuveciclib)

Atuveciclib (Table 2) was discovered by Lücking et al. in search of a highly selective, orally applicable and exclusive P-TEFb inhibitor. The basis for Atuveciclib design was the triazine BAY-958, which demonstrated in vitro IC_50_ of 11 nM against CDK9 and a CDK2/CDK9 IC_50_ ratio of 98, but slow absorption and low oral bioavailability in vivo. Atuveciclib displayed in vitro IC_50_ of 13 nM against CDK9 and an IC_50_ ratio of 100. Ex vivo IC_50_ against CDK9 were 0.92, 0.31 and 0.89 µM in HeLa, MOLM-13 and MV-14 cell lines, respectively, with excellent anti-proliferative effect. In vivo, BAY-1143572 showed markedly lower blood clearance and improved oral bioavailability. In MV-14 AML cell line derived rat xenograft, daily oral administration of 12 mg/kg of BAY-1143572 for 14 days resulted in no tumor re-growth in 9/12 animals [129]. Owing to these promising pre-clinical results, BAY-1143572 was subjected to two Phase-I studies against advanced malignancies (lymphomas and solid tumors; NCT01938638) and acute leukemia (NCT02345382). In both cases however, due to high drug related TEAE (Treatment Emergent Adverse Effect) and lack of clinal response at the dose tested, both trials were pre-maturely terminated [130,131] (Table 2).

### 4.4. BAY-1251152

BAY-1251152 (Table 2) was a follow-up, P-TEFb inhibitor of BAY-1143572, with increased potency, with IC_50_ against CDK9 under both in vitro and ex vivo (MOLM-13 cells) conditions being 3 and 29 nM, respectively; enhanced selectivity against CDK2; permeability; solubility; and no efflux. BAY-1251152 also demonstrated impressive in vivo efficacy in MOLM-13 derived xenograft mouse and rat models [132]. In our own studies, BAY-1251152 had demonstrated IC_50_ of 19.06, 16.57 and 74.64 nM against HeLa, SiHa and OVCAR-3 cells, respectively, as compared to comparative BAY-1143572 IC_50_ of 1.27 and 0.64 µM for HeLa and SiHa cells (un-published work). Until the development of AZD-4573, BAY-1251152 was the only selective CDK9 inhibitor currently undergoing clinical trials [133] against advanced malignancies (NHL and solid tumors; NCT02635672). This phase-I study revealed significant, dose-dependent reductions in the mRNA levels of *MYC*, *PCNA* and *MCL-1* and manageable safety. There were also signs of anti-tumor activity viz. stable disease (SD) in 9/31 (29%) and durable SD in 3/31 (9.7%) patients [134]. Another phase-I trial against advanced hematological malignancies (NCT02745743) is ongoing. Preliminary results from AML patients revealed significant, but short-lived post-treatment reductions in the mRNA levels of *MYC* (0.5–4 h), *PCNA* (1–4 h) and *MCL-1* (1–3 h). No objective responses were achieved at any doses, despite achieving CDK9 inhibition [135]. Additionally, BAY-1251152 monotherapy was declared to generate durable remissions of over two years in 2/7 patients with very aggressive relapsed/refractory double-hit DLBCL (Diffused Large B-Cell Lymphoma), following a phase-I trial [136] (Table 2).

### 4.5. SNS-032 (BMS-387032)

SNS-032 (Table 2), was originally shown as a selective CDK2 inhibitor, but was later shown to be able to inhibit CDK7 and CDK9, with modest inhibitory activities against CDKs 1, 4, 5 and 6. In vitro kinase profiling revealed IC_50_ values of 38–48 nM (CDK2), 62 nM (CDK7), 4 nM (CDK9), 480 nM (CDK1), 925 nM (CDK4), 340 nM (CDK5) and >1000 nM (CDK6) [137], while Albert et al. reported these values, respectively, at 6 nM (CDK2), 68 nM (CDK7), 1.4 nM (CDK9), 52 nM (CDK1), 355 nM (CDK4) and 3404 nM (CDK6) (no CDK5 values were reported) [138]. Ex vivo IC_50_ values were 231 and 192 nM against CDKs 7 and 9, respectively, in RPMI-8226 cells, along with reductions in pS5 and 2 of RNAP II, XIAP and MCL-1 levels and increased PARP cleavage [137]. In a phase-I clinical trial involving patients with advanced solid tumors, high SNS-032 concentrations inhibited pS5 and 2 of RNAP II and MCL-1 expression, in their PBMCs. However, all 20 participants of this study discontinued treatment due to relapse or disease progression (65%); deterioration without progress (20%); adverse effect (5%); drug-related toxicity (5%); and patient request (5%). Only 3 patients exhibited SD following treatment [139]. Another phase-I pharmacological study (NCT00446342) against 19 CLL and 18 MM patients, identified MTD (Maximum Tolerated Dose) and DLT of 75 and 100 mg/m^2^, respectively, for the CLL patients. MTD could not be established while DLT was not observed for the MM patients. Overall, SNS-032 was well tolerated by both patient types, at all infusion doses and induced reductions in pS5 and 2 of RNAP II, within 2 h of starting the infusion, reverting to baseline levels after 24 h. At 6 h, when the infusion was stopped, the reduction of pS2 level was more pronounced (64%) than pS5 level (35%), along with XIAP (10%) and MCL-1 (56%) levels and increased PARP cleavage. However, SNS-032 only exhibited limited anti-tumor activities in both the CLL (1 partial and 1 stable diseases) and MM cohorts (1 stable disease), with the treatment of all three stable cases terminated due to drug-related toxicities, patient request or investigator decision [140]. A third clinical trial of SNS-032 against solid tumors was also completed, although its results have not yet been posted (NCT00292864) (Table 2).

Olson et al. had recently demonstrated that conjugating a Thalidomide group to SNS-032, to produce THAL-SNS-032, could selectively and potently degrade CDK9, but no other CDK targets of SNS-032, in a CRBN dependent manner. Compared to SNS-032, THAL-SNS-032 exhibited in vitro IC_50_ values of 171 nM (CDK1), 62 nM (CDK2), 398 nM (CDK7), 4 nM (CDK9). Against a panel of leukemia cell lines, THAL-SNS-032 demonstrated anti-proliferative effects at IC_50_ values 2–6.5 folds lower than those of SNS-032. It also inhibited pS2, in MOLT4 AML cells, in a time- and dose-dependent manner, but pS5 and 7 levels remained completely unaffected. While both SNS-032 and its derivative similarly induced PARP and Caspase-3 cleavages, only THAL-SNS-032 induced MCL-1 degradation and γH2A.X activation in MOLT4 cells [141], unlike in RPMI-8226 cells [139]. Reduction in global gene expression was also significantly more pronounced after THAL-SNS-032 treatment, as compared to its predecessor [141]. In short, addition of the Thalidomide group to SNS-032, turned a pan-CDK inhibitor into a highly selective CDK9 inhibitor. Due to the CRBN-mediated ubiquitination and degradation of CDK9, the mechanisms of THAL-SNS-032-mediated induction of apoptosis and inhibition of the transcriptional machinery was also different than that of SNS-032, which only inhibited the activations of CDKs 2, 7 and 9. No clinical trials of THAL-SNS-032 has been initiated till date (Table 2).

### 4.6. AT-7519

AT-7519 (Table 2) is a pan-CDK inhibitor developed by Wyatt et al., that selectively inhibited CDKs with in vitro IC_50_ values of 190 nM (CDK1), 47 nM (CDK2), 67 nM (CDK4) and 18 nM (CDK5). It showed low oral bioavailability. Intraperitoneal injection for 8 days BID of AT-7519 in A2780 derived ovarian cancer xenograft mouse model produced 86% tumor growth inhibition at 7.5 mg/kg dose [142]. Squires et al. later identified CDK9 as a target for AT-7519 with in vitro IC_50_ values of 210 nM (CDK1), 47 nM (CDK2), 100 nM (CDK4), 13 nM (CDK5), 170 nM (CDK5) and <10 nM (CDK9). AT-7519 exhibited anti-proliferative effect against 26 cancer cell lines at IC_50_ values of 40–940 nM. 24 h of treatment of HCT-116 cells as a consequence of the inhibitions phosphorylations of substrates of CDK1 (PP1α at pT320) and CDK2 (Rb at pT821 and NPM at pT199). Global transcription was also inhibited at IC_50_ of 56 nM due to the inhibition of CDK9 mediated phosphorylation of S2 on RNAP II, within 4 h of treating HCT-116 cells with AT-7519. 24 h treatment with this inhibitor resulted in cell-cycle blockades at G0/G1 and G2/M phases, causing a reduction in S-phase and increase in G2/M phase cells and apoptosis. 9 days BID of AT-7519 in HCT116 derived colon cancer xenograft mouse model caused complete tumor regression for upto 24 days at 9.1 mg/kg dose [143]. AT-7519 also inhibited pS2 in the HL-60 cell line, reductions in MCL-1 levels and global transcription at IC_50_ of 34 nM. HL-60 xenograft model displayed inhibited pT199, pT821 for upto 16 h and MCL-1 for 24 h, and PARP cleavage, following a single dose of 10 mg/kg of AT-7519. In patient derived CLL cells, AT-7519 treatment displayed time-dependent cytotoxicity; induced apoptosis, irrespective of time and dose; inhibited pS2; reduced MCL-1 protein and mRNA; and induced PARP cleavage, in a dose-dependent manner [144]. Multiple phase I and II clinical trials have been undertaken with AT-7519 [145]. In a phase I study involving patients with advanced/metastatic solid tumors and refractory NHL (NCT00390117), 19/32 showed SD from 2.5–11.1 months, with a median duration of 3.3 months. Intravenous administration was well tolerated and served as the basis of two phase-II trials involving MCL (NCT01652144) and CLL (NCT01627054) [146]. In both these trials, although the inhibitor was well tolerated by most patients, only modest clinical activity was observed at the tested dose and schedule [147]. Another phase I study evaluating the combination of the HSP90 inhibitor-Onalespib and AT-7519 involving advanced/metastatic/unresectable solid tumors is currently active (NCT02503709) [148] (Table 2).

### 4.7. NVP-2

NVP-2 (Table 2) is a highly selective aminopyrimidine-derived CDK9 inhibitor, developed by Novartis. In vitro IC_50_ NVP-2 against CDK9 was <0.5 nM as compared to 584 nM (CDK1), 706 nM (CDK2), 1050 nM (CDK5) and >10 µM (CDK7). When used against MOLT4 cells, NVP-2 inhibited cell proliferation with an IC_50_ of 9 nM, as well as apoptosis induction after 4 h of treatment, as evidenced by PARP and Caspase-3 cleavages and complete loss of MCL-1, lasting for 24 h. Washout experiments in these cells showed a marked decrease in cell viability, accompanied by PARP and Caspase-3 cleavages and γH2A.X activation, even 72 h after NVP-2 removal. The inhibitor also reduced pS2 levels and a significant decrease in global mRNA expression, including 2-4 log2 fold reduction in CRC (Core Regulatory Circuitry) genes like *MYC*, *MYB* and *RUNX1*. ChIP-seq analysis following NVP-2 treatment revealed increased levels of promoter-proximal paused RNAP II at the TSS (Transcription Start Site), with corresponding decrease in RNAP II level all-over the gene body, including at the *MYC*, *MYB* and *RUNX1* loci. CDK9 inhibition also caused gene body wide loss of SPT5 [118,141]. These data demonstrated that NVP-2 is a small-molecule CDK9 inhibitor with tremendous clinical potential, however, no clinical trial has been reported till date (Table 2).

### 4.8. JSH-150

JSH-150 (Table 2) is a recently synthesized, highly selective CDK9 inhibitor with in vitro IC_50_ values of, determined using the Invitrogen SelectScreen platform, 1 nM (CDK9) and 1.34 µM (CDK1), 2.86 µM (CDK2), 4.64 µM (CDK5) and 1.72 µM (CDK7). JSH-150 bound to CDK9 via five H-bonds–C106 in the hinge region (2), D109 (1), D167 in the activation loop (1) and T29 in the P-loop (1). Its GI_50_ (concentration of an inhibitor causing 50% of maximum inhibition in cell proliferation) against multiple solid and hematologic cancer cell lines ranged from 1.1–44 nM as compared to 1100 nM against CHO cells. 2 h of treatment of the cell lines HL-60 and MV4-11 (AML) and MEC-1 (CLL) did not affect the phosphorylation of CDK9 at T186 and RNAP II at S5; XIAP and BCL-2 levels but inhibited pS2 (EC_50_ < 100 nM), MCL-1 and cMYC levels, in a dose-dependent manner. Treatment for 24 h induced PARP and Caspase-3 cleavages in all three cell lines, as well as cell cycle arrest at the G0/G1 phase. In vivo, T_max_ and T_1/2_ were 2 and 3.33 h; and 1.55 and 3.37 h in mice and rats, respectively. JSH-150 was also orally bio-available with bio-availabilities of 45.01 and 45.10% in mice and rats, respectively. MV4-11 derived mouse xenograft models, 20 and 30 mg/kg/day of JSH-150 treatment for 14 days, suppressed tumor progression, without causing general cytotoxicities or tumor recurrence, one week after the treatment was stopped [149]. No clinical trial of JSH-150 has been reported so far (Table 2).

### 4.9. LY-2857785

LY-2857785 (Table 2) is a reversible, ATP-competitive inhibitor, demonstrating in vitro IC_50_ values of 246 nM (CDK7), 16 nM (CDK8) and 11 nM (CDK9). Against a panel of 114 kinases, it showed IC_50_ of <100 nM against only 5 other kinases. In U2OS cells, LY-2857785 targeted pS5 and 2 of RNAP II with IC_50_ of 89 and 42 nM, respectively, and inhibited cell proliferation at IC_50_ of 76 nM. No cell-cycle arrest was detected. It also inhibited cell proliferation of the leukemic cell lines MV-4-11, RPMI-8226 and L363 at IC_50_ of 40, 200 and 500 nM, following 4–24 h of treatment, and induced apoptosis in a time-dependent manner. Levels of XIAP and MCL-1 were also reduced and triggered PARP and Caspase-3 cleavages. Proliferation of other AML cell lines were also inhibited. In HCT116-derived mouse xenograft model, LY-2857785 treatment inhibited pS2 in a dose-dependent manner with a TED_50_ of 4.4mg/kg (Threshold Effective Dose). Multiple AML xenograft models also showed dose-dependent tumor regression. Moreover, LY-2857785 showed good solubility for and pharmacokinetics. Unfortunately, it also exhibited anti-proliferative properties towards normal hematopoietic progenitor cells of humans (ex vivo), rats and dogs (ex and in vivo), causing in vivo hemetoxicity of bone marrow, gastrointestinal tract and other organs in dogs. As a result, the further clinical development of LY-2857785 was discontinued [150].

### 4.10. LDC000067

LDC000067 (Table 2) was synthesized by Albert et al. as an ATP competitive inhibitor with in vitro IC_50_ of 44 nM (CDK9), 2.4 µM (CDK2) and >10 µM (CDK6/7), as against 5.2, 15, 305 and 103 nM, respectively, for Flavopiridol [138]. As has been mentioned in the previous sections, due to the high specificity of LDC000067 against CDK9, it has proved to be a valuable tool in pre-clinical studies against multiple and diverse cancer entities [88,90,94,116]. However, no clinical trials of LDC000067 were ever undertaken (Table 2).

### 4.11. CDKI-73 (LS-007)

The novel CDK9 inhibitor CDKI-73 was the lead in class compound of a class of 5-substituted 4-(thiazol-5-yl)-2-(phenylamino) pyrimidines, designed by Wang et al., which targeted the ATP gatekeeper residue F30 and ribose-binding pocket of CDK9 [151,152]. Under in vitro conditions, CDKI-73 exhibited IC_50_ of 8.17 nM (CDK1), 3.27 nM (CDK2), 8.18 nM (CDK4), 37.68 nM (CDK6), 134.26 nM (CDK7) and 5.78 nM (CDK9) [153]. Against patient derived CLL cells, CDKI-73 was demonstrated to be significantly more potent than Flavopiridol or its analog Fludarabine. However, when these same cells were protected from the effects of cytotoxic drugs, by culturing them with CD40L-expressing MEFs, CDKI-73 retained its cytotoxic effects while that of Fludarabine was completely annulled. It was equally effective against these primary CLL cells (n = 28) as against those derived from clinically relapsed CLL patients (n = 10), including those harboring p53 deletion (n = 3) while exhibiting superior bioavailability than Flavopiridol. Once again, treatment of the primary CLL cells for 4 h with 0.1 µM of CDKI-73 inhibited pS2 and T186 phosphorylations of RNAP II and CDK9, respectively, as well as depleted MCL-1 expression. Interestingly, CDKI-73 demonstrated synergistic cytotoxic effects when combined with Fludarabine, against primary CLL cells, cultured under both normal and cytoprotective conditions [152]. Significantly, CDKI-73 exhibited LD_50_ (median dose of an inhibitor required to kill 50% of a population within a specified time) values of only 0.08 µM (primary CLL cells) as opposed to 40.5 µM (normal B-cells) and 23.0 µM (normal CD34^+^ bone marrow cells), as compared to LD_50_ of 0.35, 0.59 and 0.52 µM for the respective cell types, in the presence of Flavopiridol, demonstrating the excellent safety potential of CDKI-73 [152]. Against AML and ALL cell lines and primary patient derived cells, CDKI-73 and Flavopiridol showed very similar IC_50_ (163 nM vs. 147 nM) and LD_50_ [67 nM vs. 100 nM (ALL) and 102 nM vs. 112 nM (AML)] values. Ex vivo CDKI-73 inhibited pS2 and pS5 phosphorylations of RNAP II and pT320 phosphorylation of PP1α at, in a dose-dependent manner, albeit the highest inhibition of pS2 levels, in line with its in vitro IC_50_ values. CDKI-73 also induced dose- and time-dependent apoptosis in AML and ALL cell lines and patient derived primary cells, as demonstrated by enhanced PARP and Caspase-3 cleavages and reduced expressions of XIAP and MCL-1 protein and mRNA [153].

Interestingly, in their recent work, Sorvina et al. had demonstrated a novel effect of CDKI-73 in causing the accumulation of large multi-vesicular Rab11 endosomes close to the cell periphery, thereby inhibiting the delivery of the Rab11 vesicles to the plasma membrane. This prevented the trafficking and secretion of anti-microbial peptides like Drosomycin, pro-inflammatory cytokines like IL-6 and TNFα during an innate immune response. Thus, CDKI-73 could be a potent anti-inflammatory therapy against unwarranted inflammatory response in patients/people with inflammatory disorders [154]. Till date, no clinical studies have been undertaken with CDKI-73.

**Table 2 cancers-13-02181-t002:** The various CDK9 inhibitors, their target kinase (s), cancer entities they were tested against (includes pre-clinical trials where clinical trials were never performed) and their clinical trials (wherever performed). Source: www.clinicaltrials.gov (accessed on 12 February 2021).

Inhibitor	CDKs	Against	Clinical Trial and Status	Ref.
Flavopiridol (Alvocidib) 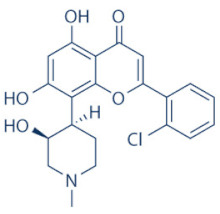	1, 2, 4, 6, 7 and 9	Multiple cancer entities	Numerous; three active trials (NCT03604783; NCT03969420; NCT03593915)	
SCH-727965 (Dinaciclib) 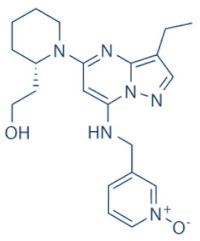	1, 2, 5 and 9	Multiple cancer entities	Numerous; four active trials (NCT01676753; NCT03484520; NCT01434316; NCT00937937)	[123,124]
LDC000067 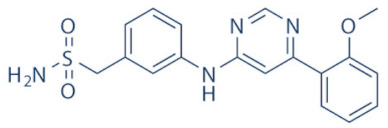	2 and 9	Multiple cancer entities (pre-clinical studies)	None	[138]
BAY-1143572 (Atuveciclib) 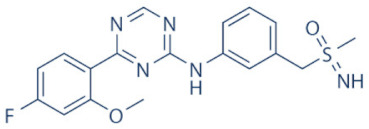	9	Acute leukaemia and advanced malignancies	NCT02345382, NCT01938638 (complete/pre-maturely terminated)	[130,131]
BAY-1251152 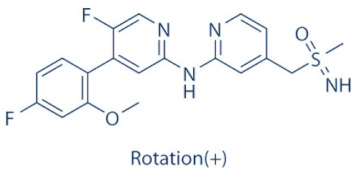	9	Advanced hematological cancers and advanced malignancies	NCT02745743 (complete), NCT02635672 (active)	[132,134,135]
SNS-032 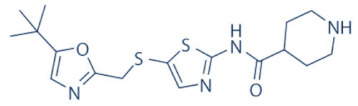	2, 7 and 9	Advanced solid tumors and advanced B-lymphoid malignancies	NCT00446342, NCT00292864 (complete)	[137,139,140]
THAL-SNS-032 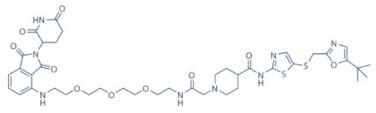	9	ALL (Acute Lymphoblastic Leukemia) (pre-clinical studies)	None	[141]
AZD-4573 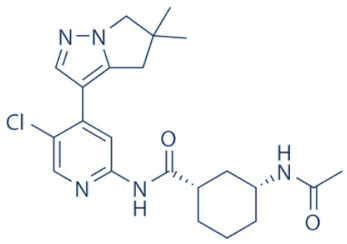	9	Advanced hematological cancers and relapsed/refractory hematological cancers	NCT04630756, NCT03263637 (active)	[125,126,127,128]
NVP-2 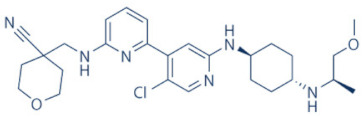	9	ALL (pre-clinical studies)	None	[118,141]
JSH-150 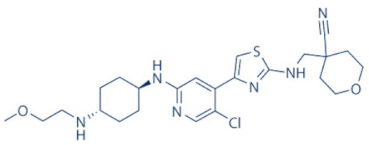	9	AML and CLL (pre-clinical studies)	None	[149]
LY-2857785 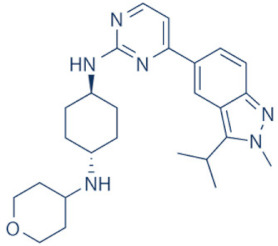	7 and 9	Multiple cancer entities (pre-clinical studies)	None	[150]
AT-7519 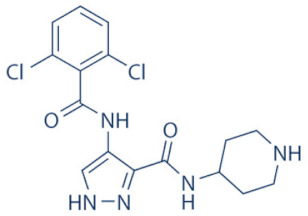	1, 2, 4, 5 and 9	Advanced/metastatic/unresectable solid tumors, refractory NHL, MM, MCL, CLL	NCT02503709 (active), NCT00390117, NCT01183949, NCT01652144, NCT01627054 (complete)	[144,145,146,148]
CYC-202 (Roscovitine) 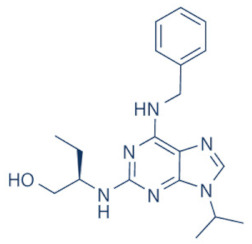	1, 2, 4, 5, 7 and 9	TNBC, NSCLC, advanced solid tumors, Cushings disease	NCT01333423 (withdrawn), NCT00372073 (terminated), NCT00999401, NCT02160730 (terminated), NCT03774446 (recruiting)	[155,156]
CR-8 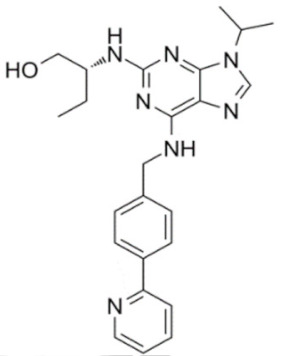	1, 2, 4, 5, 7 and 9	Neuroblastoma, (pre-clinical studies)	None	[155,156]
CDKI-73 (LS-007) 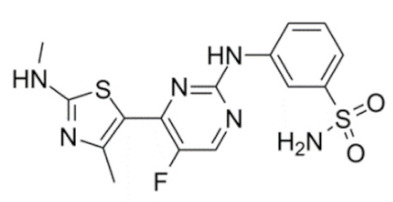	1, 2, 4 and 9	CLL, AML, ovarian cancer (pre-clinical studies)	None	[152,153]

## 5. Conclusions

CDK9 is an extremely critical kinase in regulating the productive transcription of several anti-apoptotic and oncogenic genes, essential for the maintenance, growth, metastasis and chemo-resistance of cancers. In this review, we have only touched upon the proverbial tip of this iceberg. Apart from the cancer entities described here, CDK9 is also involved in many other cancer entities like—pediatric Soft Tissue Sarcomas (STS) like Rhabdomyosarcoma (RMS), Ewing’s Sarcoma (ES), Synovial Sarcoma (SS) and Malignant Rhabdoid Tumors (MRT) [1]; Neuroblastoma; Primary Neuroectodermal Tumor (PNET) [157]; cervical cancer [158]; CLL [159]; Glioblastoma [160] etc. The functions of CDK9 is not just limited to cancers, but in other diseases as well like Rheumatoid arthritis [161]; cardiomyocytes development [162]; T-cell activation [163]; replication of viruses like HIV-1 and -2, EBV, CMV, HSV-1 and -2, HTLV-1 [95,164] etc. A great of interest and effort has therefore been invested in targeting the activation of CDK9 or promoting its degradation. As mentioned in our review, many of these inhibitors were put through clinical trials, Flavopiridol being the first of them [149]. Unfortunately, while most of these inhibitors showed promise during pre-clinical trials, many elicited severe side effects or were not effective in generating SD or improving the OS and PFS in patients. Several factors can be attributed to these disappointing outcomes like the complex and challenging toxicology profiles of the pan-CDK inhibitors (Table 2) [150] and the critical roles played by CDK9 in the development of normal, surrounding cells. The low numbers of patients with mutations or amplifications in *CDK9* for most cancers (Figure 5), a key feature that helped in selectively targeting other kinases like BRAF (Vemurafenib and Dabrafenib against BRAF_V600E_) and MEK1/2 (Trametinib against BRAF_V600E/K_) specifically in cancer cells [165,166], probably also affected the effectiveness of the next generation inhibitors, which targeted CDK9 far more selectively. Encouragingly enough, during their phase-I clinical trials the 3rd generation inhibitors-BAY-1251152 and AT-7519 had generated generally favorable outcomes involving AML, DLBCL, solid tumors and refractory NHL. AT-7519 was also subjected to two phase-II trials against MCL and CLL [136]. The latest inhibitor—AZD-4573 is presently undergoing two phase-I trials against advanced hematological cancers [127,128]. These promising recent developments guarantee that the endeavors to target CDK9 activity would undoubtedly continue to be pursued and further refined with newer inhibitors and other strategies, as suggested below.

Thus far, most of the clinical trials undertaken with the 2nd and 3rd generation CDK9 inhibitors had been as monotherapies (Table 2). However, one possible way to improve the clinical relevance of these inhibitors is to take advantage of the pathways that regulate the activity of CDK9 and the pathways it in-turn regulates. As explained earlier, BRD4 recruits CDK9 to the promoters [75]. The up-regulation of BRD4 expression is also intimately related with carcinogenesis and poor prognosis [167]. Additionally, unlike CDK9, the genetic alterations of BRD4 also occur with increasing frequencies in different cancer entities (Figure 6A). These reasons have led to the development of a number of inhibitors against BRD4/BET family members [75,167]. Recently, McCalmont et al. had demonstrated the *ex* and in vivo effects of combining a CDK9 inhibitor and two BET inhibitors. Combinations of CDKI-73 with JQ1 or iBET-151 resulted in the synergistic reduction in cell viability, improved median EFS (Event-Free Survival), as compared to monotherapies against infant ALL and adult AML PDX models harboring the *MLL*-rearrangement (*MLL*-r) gene. Significant reduction of leukemic burdens in the bone marrow and spleen were also observed, along with reductions in MYC and BCL-2 protein levels. pS2 levels were also reduced by 49% with the combination against 29% with monotherapy. Transcripts of known AML promoting genes-*PIM1*, *HLX*, *TRAF6* and *TRIB3* were also reduced. Only the combination therapy induced significant negative enrichment of transcription factor genes associated with super-enhancers, in the MLL cell line K562 [168]. Another BET inhibitor BI-894999, was also reported to be strongly effective against AML cell lines, patient derived primary materials, and mouse models, more so in combination with LDC000067 in reducing pS2 levels and enhancing apoptosis. Noteworthy, BI-894999 is a part of an active phase 1a/b trial against patients with advanced hematological cancers (NCT02516553) [169]. Additionally, combing LDC0000167 with the BRD4 inhibitors-JQ1 or iBET-762 against malignant rhabdoid tumors, down-regulated the anti-apoptotic genes *MCL-*1, *BCL-*6 and *BTG1* as well as *MYC* and inhibited cell-proliferation and tumor growth in vitro and in vivo [170]. Several other BRD4/BET inhibitors like ABBV-075 [171], PLX-51107 [172], CC-90010 and ZEB-3694 are undergoing clinical trials [173,174]. Combining BRD4 inhibitors could potentially improve the efficacy of CDK9 inhibitors.

The other reservoir of active P-TEFb–SEC (Figure 3E) has also been targeted recently using the specific inhibitors KL-1 and its structural homolog KL-2, which interfered with association of Cyclin T1 of P-TEFb with the AFF4 subunit of SEC to disrupt the progress of transcription elongation after paused transcription initiation. SEC plays an important role in the transcription of the *MYC* gene. Likewise, treating the cMYC over-expressing lung cancer cell line H-2171 with either of these two inhibitors resulted in a dose-dependent down-regulation of cell proliferation. In vivo, treating the cMYC-dependent tumor mouse model MDA-231-LM2 with KL-1 or -2 significantly delayed tumor growth and improved the survival of the mice. Apoptosis of the MDA-231-LM2 cells also increased in the presence of the inhibitors [175,176]. As mentioned earlier, SEC recruitment required the pre-occupancy of BRD4 [64,65]. Therefore, inhibiting CDK9, BRD4 and SEC in combination could prove to be a highly effective trident of inhibitors against cancers, especially against the cMYC driven ones.

Combination of the HSP90 inhibitor Onalespib with the AT-7519 is also undergoing a clinical trial [148]. Other clinical trials of HSP90 inhibitors are also undergoing trials [177] and could be potential combinations with CDK9 inhibitors. As evident from the previous sections, almost all CDK9 inhibitors reduced MCL-1 levels. Therefore, combination of CDK9 inhibitors with those against MCL-1 could also significantly improve the effect of the CDK9 inhibitors. Several such inhibitors are undergoing clinical trials and have been reviewed elsewhere [178,179]. Other potential combinations could be with inhibitors of XIAP [180] or BCL-2 [181]. Xie et al. had recently reported that the potent, selective and orally bioavailable BCL-2 inhibitor ABT-199 (Venetoclax), in combination with the potent CDK9 inhibitor CDKI-73 (LS-007) synergistically induced apoptosis in the acute leukemia (AML and ALL) cell lines HL-60, CCRF-CEM and Molt-4, accompanied by increased PARP and Caspase-3 cleavages and reduction in XIAP and MCL-1 expression [153]. Lu et al. had recently reported that inhibiting CDK9 activity with their newly developed inhibitor i-CDK9 suppressed phosphorylations of RNAP II at S2 and SPT5 at T775, induced genome-wide pausing of RNAP II at gene promoters. Paradoxically, the expressions of *MYC* and other primary response genes went-up upon sustained i-CDK9 treatment, a potential mechanism of acquired resistance. They had proposed combing the inhibitors of CDK9 with those against BRD4 and cMYC to efficiently inhibit the proliferation and promote apoptosis of cancer cells [182]. *MYC* is also one of the most frequently altered oncogenes (Figure 6B), being aberrantly expressed in <70% of all cancers. Even though initially deemed a difficult target to inhibit, more promising inhibitory strategies against cMYC have been recently developed [183], and could serve as potential combination partners with CDK9 inhibitors. The most promising cancer entities to benefit from inhibiting CDK9, or its up-stream regulators or its down-stream targets would probably be those of leukemic origin as they have clearly defined genetic re-arrangements (like *MLL*-r) [184] or oncogenic dependence (like *MYC*) [185,186], allowing these cells to be targeted more specifically. Many of the pre- and clinical studies with the CDK9 inhibitors have also been undertaken towards patients with different forms of leukemia (Table 2).

In summary, newer generations of CDK9 inhibitors are already opening-up its untapped potential as an anti-cancer therapy and ongoing works in this direction are helping develop better, more selective inhibitors. However, it would be prudent that these CDK9 inhibitors are combined with inhibitors of other proteins/oncogenes/inhibitors of apoptosis/protein complexes like BRD4, SEC, HSP90, cMYC and MCL-1. This would potentially help in— (1) delaying/avoiding potential therapeutic resistance; (2) enhancing the diversity of the treatment regimen; and (3) improving the efficacy of the CDK9 inhibitors. All the above-mentioned inhibitors or inhibition strategies also require to be able to distinguish between normal rapidly proliferating cells like the T-cells and cancer cells. CDKI-73, for example has been reported to have much lower toxicity against T- and B-cells as compared to CLL cells [152,153]. Wogonin, a natural plant-based flavone was shown to target CDK9 activity, in an ATP-competitive manner, resulting in lower pS2 and MCL-1 levels and increased apoptosis in CLL and multiple other cancer entities. However, more importantly, Wogonin preferably induced apoptosis in cancer cells over normal cells. Additionally, like CDKI-73, Wogonin and related flavones like Baicalein, Apigenin, Chrysin and Luteolin enhanced the activity of the specific BCL-2 inhibitor ABT-263 in primary AML and ALL cells and cell lines of different cancer entities, while also sensitizing cancer cells harboring acquired ABT-263 resistance [187,188]. More exacting pre-clinical trials are necessary to better understand the nuances in the functions of CDK9 between normal and cancer cells, to be able to specifically target the latter.

## Figures and Tables

**Figure 1 cancers-13-02181-f001:**
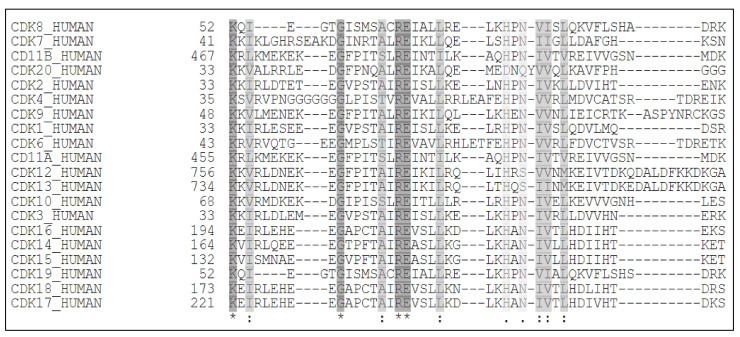
The PITALRE (Pro-Ile-Thr-Ala-Lue-Arg-Glu) sequence of CDK9 which aligned with the highly conserved PSTAIRE box, observed in multiple CDKs.

**Figure 2 cancers-13-02181-f002:**
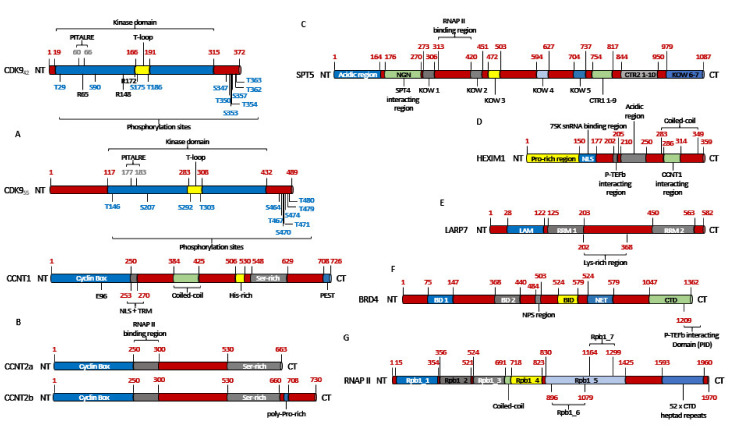
The domain structures of key proteins involved in P-TEFb regulated transcription. (**A**) The two CDK9 isoforms expressed in most cells. The CDK9_55_ is longer than the CDK9_42_ by 117 amino acids (aa) at the N-terminus (NT) as the former isoform is transcribed by a different promoter, <500 bp away from the later, on the CDK9 gene. The aa highlighted in blue represent the key phosphorylation sites on CDK9. The PITALRE region is marked in light black. While the three acetylation sites are highlighted in black (**B**) The three main cyclins that partner with CDK9. The main acetylation site E96 is highlighted in black on Cyclin T1 (**C**) SPT5 which is a component of the 5,6-dichloro-1-β-D-ribofuranosylbenzimidazole (DRB) Sensitivity-Inducing Factor (DSIF). It possesses upto 7 Kyrpides-Ouzounis-Woese (KOW) domains which binds to different regions of RNAP II. The RNAP II binding region of SPT5 has been highlighted in black. (**D**) HEXIM1. (**E**) LARP7 with its La-Motif (LAM) and RNA-Recognition Motifs (RRM1 and 2). The LAM and RRM1 motifs work synergistically to form a ‘V’ shaped clamp to bind to the 3′-UUUU-OH region of 7SK snRNA while the RRM2 binds to the apical loop of the 3′-hairpin of the 7SK snRNA. (**F**) BRD4 with its N-terminus Phosphorylation Site (NPS), two Bromodomains (BD1 and 2) and N-terminus Extra Terminal (NET) domain and the C-terminus P-TEFb Interacting Domain (PID). (**G**) RNAP II with its 7 Rpb1 (RNA polymerase II subunit B1) domains and the 52 Y_1_S_2_P_3_T_4_S_5_P_6_S_7_ heptad tandem repeats at its CTD.

**Figure 3 cancers-13-02181-f003:**
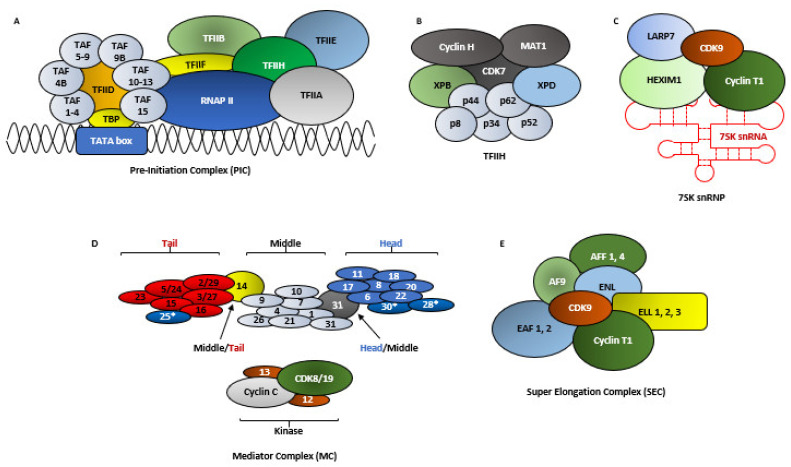
The structures of key complexes involved in RNAP II mediated transcription and regulation of CDK9 activity. (**A**) Pre-Initiation Complex (PIC)–the PIC is composed of RNAP II which is assisted by General Transcription Factors (GTFs) like TFIIA, TFIIB, TFIID, TFIIE, TFIIF and TFIIH. TFIID in-turn is composed of the TATA box-Binding Protein (TBP) and upto 15 TBP Associated Factors (TAF). TFIID, assisted by TFIIA, first recognizes the core promoter by scanning for and subsequently associating with the TATA box sequence on the promoter DNA. TBP then recruits the TFIIB which subsequently loads the RNAP II-TFIIF complex on the promoter. Eventually, TFIIE and TFIIH facilitates the opening of the transcription bubble [62]. (**B**) The TFIIH is a heterodecameric protein complex comprising of the proteins XPB, XPD, p62, p52, p44, p34 and p8 forming the core, plus CAK with CDK7, Cyclin H and MAT1. It is an essential part of PIC and is also involved in DNA repair. The DNA helicase XPD and the dsDNA translocase activity of XPB are both required for opening of the transcription bubble, although XPD is not critical for transcription initiation but for DNA damage repair. CAK (CDK7/Cyclin H) phosphorylates S5 of RNAP II, whereas MAT1 both assists in the interaction between CDK7 and Cyclin H and recruits CAK to the TFIIH core, by interacting with XPD and XPB. The function of XPB is regulated by p52 and p8 while that of XPD is regulated by p44 [46]. (**C**) The 7SK snRNP is composed of the non-coding 7SK snRNA, serving as as scaffold for the RNA binding proteins HEXIM1/HEXIM2, LARP7 and MePCE. (**D**) The Mediator Complex (MC) is comprised of upto 30 subunits in humans, which are generally divided into the head, middle, tail and kinase modules. The head is composed of the mediators 6, 8, 11, 17, 18, 20 and 22 in the head (**blue**); 1, 4, 7, 9, 10, 21 and 31 in the middle (**black**); and 15, 16, 23, 2/29, 3/27 and 5/24 in the tail (**red**). The mediators 31 and 14 connect the head with middle and tail with middle, respectively. The kinase domain is composed of the mediators 12 and 13, CDK8 or its paralogue CDK19 and Cyclin C. The functions of the mediators 25, 28 and 30 (*) are yet undefined. The MC promotes the assembly of the PIC by serving as a functional bridge between RNAP II and the above mentioned GTFs [63]. (**E**) The Super Elongation Complex (SEC) is composed of CDK9/Cyclin T1, the Eleven-nineteen Lysine-rich Leukemia (ELL) proteins 1, 2 and 3, the AF4/FMR2 Family members 1 and 4 (AFF 1 and 4), Eleven-Nineteen Leukemia (ENL), ALL1-Fused gene from chromosome 9 (AF9) and the ELL-associated factors 1 and 2 (EAF 1 and 2) [64,65].

**Figure 4 cancers-13-02181-f004:**
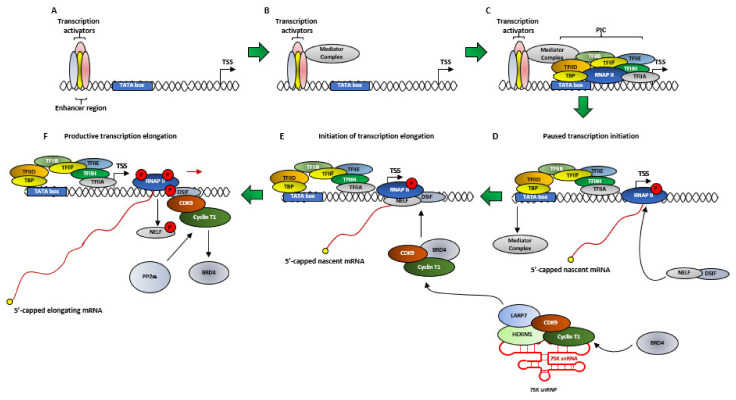
The simplified model of the regulation of transcription by P-TEFb. (**A**) At the beginning of transcription, Transcription Factors (TFs) bind to the enhancer regions, upstream of the core promoter region (represented by the TATA box). The TATA box in-turn is located upstream of the Transcription Start Site (TSS). (**B**) TFs recruit the Mediator Complex (MC) (Simplified here from Figure 3). (**C**) The MC then assists in the recruitment and assembly of the PIC on the DNA strand which, as mentioned before, starts promoter melting and form the transcription bubble, to allow to the RNAP II to access the DNA template strand. (**D**) The TFIIH subunit of PIC also phosphorylates the S5 residue at the CTD of RNAP II to initiate transcription from the TSS to generate a nascent mRNA transcript of ~50 ribonucleotides but pauses due to recruitment of two negative regulators of transcription DSIF and NELF. The S5 phosphorylation also enables the dissociation of the MC and freeing-up of the RNAP II from the PIC and promotes the recruitment of capping enzymes to cap the 5′-end of the nascent mRNA strand in a multi-step process, to prevent the nascent strand from being degraded by nucleases like XRN2. The cap remains until transcription finishes and the mRNA is properly processed. (**E**) When it is ideal for the cells to carry-out productive transcription, BRD4 recruits P-TEFb (CDK9/Cyclin T1) from its negative regulatory complex—7SK snRNP to the RNAP II. However, the kinase activity of the BRD4 bound CDK9 remains transiently inhibited due to the phosphorylation of T29 of CDK9 by BRD4 (**F**) The phosphatase PP2α is then recruited which dephosphorylates T29, restoring the kinase activity of CDK9. This causes the additional P-TEFb mediated phosphorylations of RNAP II at S2, the SPT5 subunit of DSIF at T4 and NELF. This allows NELF to dissociate from RNAP II, which along with SPT5 phosphorylation, transforms DSIF to a positive elongation factor. These events relieve RNAP II from transcription pause and allows it to progress along with the DNA template to initiate productive transcriptional elongation. As the transcription elongation gears-up, BRD4 was released from the P-TEFb.

**Figure 5 cancers-13-02181-f005:**
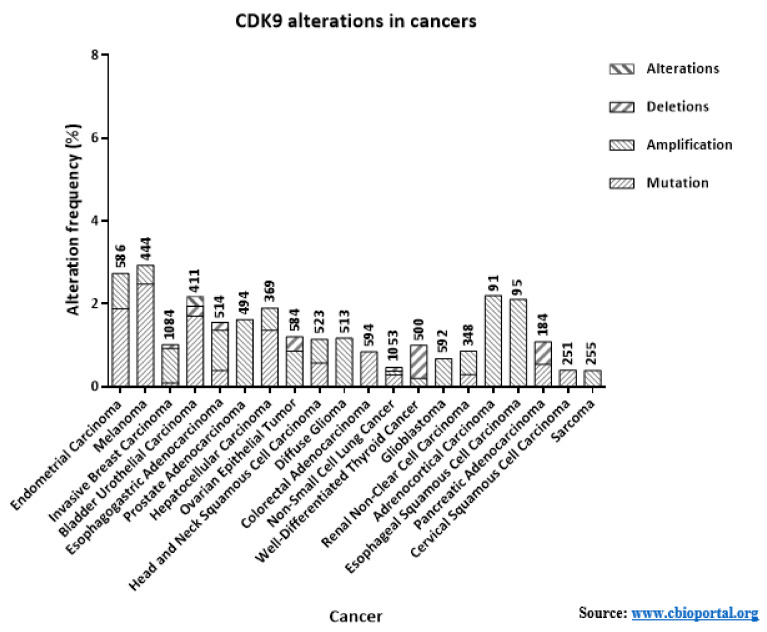
The alterations of the *CDK9* gene in various cancer entities. The alteration frequency of the *CDK9* gene in various cancers have been represented graphically. The alterations involve deletions, mutations, copy number amplifications and other alterations. The numbers on top of each cancer entity are the respective patient numbers represented. The data were obtained from www.cbioportal.org (accessed on 16 February 2021).

**Figure 6 cancers-13-02181-f006:**
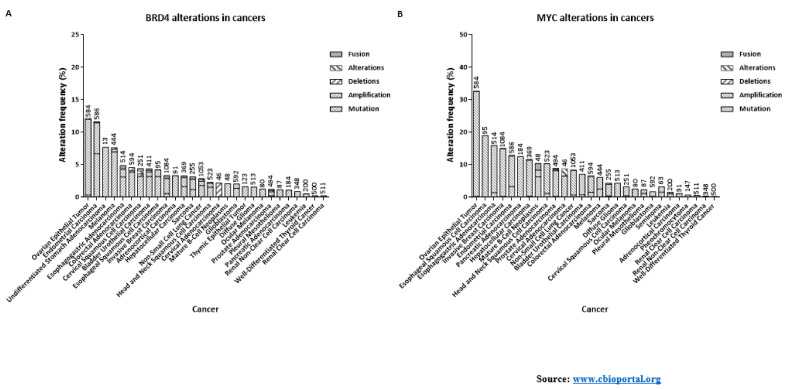
The alterations of the (**A**) *BRD4* and (**B**) *MYC* genes in various cancer entities. The alteration frequency of the *BRD4* and *MYC* genes in various cancers have been represented graphically. The alterations involve deletions, mutations, copy number amplifications, other alterations and fusions. The numbers on top of each cancer entity are the respective patient numbers represented. The data were obtained from www.cbioportal.org. (Accessed on 21 February 2021).

**Table 1 cancers-13-02181-t001:** The different known members of the CDK family, their regulating cyclin partners (except CDK5) and their general functions.

	Cyclin/Regulating Partners	Functions	Reference
CDK1	Cyclin B1	Cell-cycle regulation—promotes G2/M transition, regulates G1 progress and G1/S transition	[4,5]
CDK2	Cyclins A/D1/E1	Cell-cycle regulation—G1/S transition, exit from S-phase; initiation of DNA synthesis	[6]
CDK4	Cyclin D	Cell-cycle regulation—G1-phase transition; partial phosphorylation of Rb with CDK6	[7]
CDK5	p35/p39	All aspects of neuronal physiology; immune response; angiogenesis; myogenesis; melanogenesis and regulation of insulin levels	[6,8,9]
CDK6	Cyclin D	Cell-cycle regulation—G1-phase transition; partial phosphorylation of Rb with CDK4	[7]
CDK7	Cyclin H	CDK Activating Kinase (CAK)—phosphorylates cell-cycle regulating kinases; transcription regulation—S5 phosphorylation on RNAP II CTD to initiate transcription initiation	[10]
CDK8	Cyclin C/Med12/Med13	Part of Mediator Complex (MC), regulates the phosphorylation transcription factors, their activity and turn-over	[11]
CDK9	Cyclins T1/T2a/T2b	Positive regulation of transcription elongation	[12]
CDK10	Cyclin M	Cell-cycle regulation and tumor suppressor	[13,14]
CDK12	Cyclin K	Positive regulation of transcription elongation	[15,16]
CDK13	Cyclin K	Positive regulation of transcription elongation	[15,17]
CDK14	Cyclin Y	Regulation of cell-cycle, proliferation, migration and invasion	[18]
CDK15	Unknown	Inhibits apoptosis by phosphorylating Survivin on T34	[19]
CDK16	Cyclin Y	Promotes proliferation in medulloblastoma, prostate, breast, melanoma and cervical cancers, inhibits apoptosis by down-regulating the tumor suppressor p27 in NSCLC	[20,21,22]
CDK17	Unknown	Down-regulation causes poor prognosis in glioma. Unknown functions	[23]
CDK18	Cyclins A2 and E	Negative regulator of cell migration and adhesion, prevents the accumulation of DNA damage and genome instability	[24,25,26]
CDK19	Cyclin C	CDK8 homolog, part of Mediator Complex (MC), promotes proliferation and mitotic gene expression in the absence of CDK8 expression, negative regulation of NOTCH signaling	[11]
CDK20	Cyclin H and CK2 (generic CDK20); KCNIP2 and SNAPIN (cardiac CDK20)	Cell-cycle regulator (generic CDK20) and promotes cell survival (cardiac CDK20)	[27]
CDK21	Unknown	Regulates spermatogonial proliferation and meiosis progression and germ line cell activation in testis; unknown function in cancer	[28]

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
