# Peer review of "Targeting CDK9 for Anti-Cancer Therapeutics"

_cancers, 2021, doi:10.3390/cancers13092181_

Round 1
Reviewer 1 Report
This is a nicely written, comprehensive review. References are adequate and combine both a good historical and clinical perspective.
Minor comments
The simple Summary needs to be completed.
For the cancer relevance section, please not which types of cancer are being referenced. For example, the lung cancer section does not specify between adeno, squamous, or small cell lung cancer.
Author Response
Response to Reviewer 1 Comments
Dear reviewer 1,
Thank you very much for your helpful comments. In the following paragraphs, we have addressed both the points that you had raised.
Minor Point 1: The simple Summary needs to be completed.
Response 1: We have included Simple Summary of our review article in the revised manuscript
Minor Point 2: For the cancer relevance section, please not which types of cancer are being referenced. For example, the lung cancer section does not specify between adeno, squamous, or small cell lung cancer.
Response 2: We had specified, wherever possible, the cancer sub-type, when discussing the clinical relevance of CDK9 in a particular cancer entity, in detail. For example, in the original manuscript, under lung cancer, we had mentioned lung adenocarcinoma and NSCLC; under breast cancer, we had mentioned specifically about ERa+ breast cancer; under AML, we had mentioned about R/R AML cancer; and under melanoma, we had mentioned uveal melanoma.
In the revised manuscript, we have included the sub-types of the referenced endometrial cancer patient samples; type of melanoma cell lines used by the Zhang group (Reference 84); sub-types of the referenced ovarian cancer patient samples; and the androgen responsiveness of the prostate cancer cell lines used by the Chen group (Reference 78).
We hope that our revisions satisfactorily resolve your concerns in our original manuscript.
Sincerely yours,
Klaus Strebhardt, PhD.
Prof. of Cancer Biology

Reviewer 2 Report
This is a comprehensive, concise and meticulous review of both, basic and clinical biology of CDK9 as a therapeutic target in cancer. It puts CDK9 in the context of the CDK family, describes structure and activation of CDK9, it critical role in RNA Polymerase II-mediated transcription, the regulation of CDK9 activity by interaction with BRD4 as well as alternate activitation mechanisms and finally the clinical relevance and the inhibitors of CDK9.
Major:
(1) Since CDK9 is extremely critical in regulating the transcription of many genes, essential for NORMAL development, maintenance and growth, severe on-target side effects are anticipated. Thus, the role of CDK9, BRD4, P-TEFb and RNAP II in normal cell proliferation, differentiation as well as in the development of an organism deserves some more attention in the discussion to discriminate cancer targets from targets in normal rapidly proliferating cells, e.g. T cells and to discern anti-cancer from anti-inflammatory effects. It should also be commented on to which degree this is possible or not.
(2)CDKI-73 (Synonym: LS-007), an orally active and highly efficacious CDK9 inhibitor, with Ki values of 4 nM, 4 nM and 3 nM for CDK9, CDK1 and CDK2, could be included in Tab. 2 and section 4. Besides of down-regulating RNAPII phosphorylation, CDKI-73 is also a novel pharmacological inhibitor of Rab11 cargo delivery and innate immune secretion, fulfilling the authors' postulate that CDK9 inhibition is to be combined with inhibition of other proteins.
Minor:
The figures are cut off at the right margins.
Simple Summary written layman’s terms is missing.
I suggest to write e.g did not instead of didn't.
A ? is missing in line 352 after Cyclin T1.
restinant is a misspelling in line 387.
Author Response
Response to Reviewer 2 Comments
Dear reviewer 2,
Thank you very much for your helpful comments. In the following paragraphs, we have addressed all the major and minor points that you had raised.
Major Point 1: Since CDK9 is extremely critical in regulating the transcription of many genes, essential for NORMAL development, maintenance and growth, severe on-target side effects are anticipated. Thus, the role of CDK9, BRD4, P-TEFb and RNAP II in normal cell proliferation, differentiation as well as in the development of an organism deserves some more attention in the discussion to discriminate cancer targets from targets in normal rapidly proliferating cells, e.g. T cells and to discern anti-cancer from anti-inflammatory effects. It should also be commented on to which degree this is possible or not.
Response 1: We have added some additional statements in the discussion of the revised manuscript to address this concern
Major Point 2: CDKI-73 (Synonym: LS-007), an orally active and highly efficacious CDK9 inhibitor, with Ki values of 4 nM, 4 nM and 3 nM for CDK9, CDK1 and CDK2, could be included in Tab. 2 and section 4. Besides of down-regulating RNAPII phosphorylation, CDKI-73 is also a novel pharmacological inhibitor of Rab11 cargo delivery and innate immune secretion, fulfilling the authors' postulate that CDK9 inhibition is to be combined with inhibition of other proteins.
Response 2: We have included information about CDKI-73 in sections 4.11. Discussion and Table 2, in the revised manuscript
Minor Point 1: The figures are cut off at the right margins.
Response 1: We have corrected all the figures from “widescreen” to “A4” size, in the revised manuscript
Minor Point 2: Simple Summary written layman’s terms is missing.
Response 2: We have included Simple Summary of our review article in the revised manuscript
Minor Point 3: I suggest to write e.g did not instead of didn't.
Response 2: All mentions of didn’t have been replaced with did not in the revised manuscript
Minor Point 4: A ? is missing in line 352 after Cyclin T1.
Response 1: This punctuation has been corrected in the revised manuscript
Minor Point 5: restinant is a misspelling in line 387.
Response 2: This word has been corrected in the revised manuscript
We hope that our revisions satisfactorily resolve your concerns in our original manuscript.
Sincerely yours,
Klaus Strebhardt, PhD.
Prof. of Cancer Biology
